# Dose Optimization of Intravenous Indocyanine Green for Malignant Lung Tumor Localization

**DOI:** 10.3390/jcm13102807

**Published:** 2024-05-10

**Authors:** Hideki Ujiie, Ryohei Chiba, Akihiro Sasaki, Shunsuke Nomura, Haruhiko Shiiya, Shohei Otsuka, Hiroshi Yamasaki, Aki Fujiwara-Kuroda, Kazuto Ohtaka, Masato Aragaki, Kazufumi Okada, Yuma Ebihara, Tatsuya Kato

**Affiliations:** 1Department of Thoracic Surgery, Hokkaido University Hospital, Sapporo 060-8648, Japankatotatu7@msn.com (T.K.); 2Data Science Center, Promotion Unit, Institute of Health Science Innovation for Medical Care, Hokkaido University Hospital, Sapporo 060-8648, Japan; 3Department of Gastroenterological Surgery II, Faculty of Medicine, Hokkaido University, Sapporo 060-8638, Japan

**Keywords:** indocyanine green (ICG), fluorescence spectroscopy, optimal dose, surgical outcomes, lung tumors

## Abstract

**Background:** Intravenously administered indocyanine green (ICG) accumulates in lung tumors, facilitating their detection via a fluorescence spectrum measurement. This method aids in identifying tumor locations that are invisible to the naked eye. We aim to determine the optimal ICG dose and administration method for accurate tumor identification during lung resection surgeries, utilizing a novel ICG fluorescence spectroscopy system for precise tumor localization. **Materials and Methods:** ICG should be dissolved in the provided solution or distilled water and administered intravenously approximately 24 h before surgery, beginning with an initial dose of 0.5 mg/kg. If the tumor detection rate is insufficient, the dose may be gradually increased to a maximum of 5.0 mg/kg to determine the optimal dosage for effective tumor detection. This fluorescence spectroscopy during surgery may reveal additional lesions that remain undetected in preoperative assessments. The primary endpoint includes the correct diagnostic rate of tumor localization. The secondary endpoints include the measurement of the intraoperative ICG fluorescence spectral intensity in lung tumors, the assessment of the operability and safety of intraperitoneal ICG administrations, the measurement of the ICG fluorescence spectral intensity in surgical specimens, the comparison of the spectral intensity in lung tissues during collapse and expansion, the correlation between ICG camera images and fluorescence spectral intensity, and the comparison of fluorescence analysis results with histopathological findings. The trial has been registered in the jRCT Clinical Trials Registry under the code jRCTs011230037. **Results and Conclusions:** This trial aims to establish an effective methodology for localizing and diagnosing malignant lung tumors, thereby potentially improving surgical outcomes and refining treatment protocols.

## 1. Introduction

Lung cancer is the leading cause of cancer-related deaths worldwide. The advent of widespread computed tomography (CT) screening has significantly increased the detection rate of lung cancer, enabling its early intervention and potentially improving patient outcomes [1]. Traditionally, the standard treatment for lung cancer involves lobectomy and lymph node dissection. However, for small and early stage tumors, partial resection is increasingly preferred to preserve lung function [2].

Accurate localization is particularly challenging in minimally invasive procedures, such as video-assisted thoracic surgery (VATS), where the ability to palpate the tumor is limited. This challenge is further exacerbated in robot-assisted thoracic surgery and uniportal VATS.

Various techniques have been reported for localizing lung tumors during thoracoscopic surgery. Preoperative methods include CT-guided percutaneous localization [3], transbronchial localization, ultrasound-guided localization, and 3D-CT-guided localization [4]. Recently, virtual-assisted lung mapping has emerged for clinical application [5]. Intraoperative approaches include the staining and marking of lung tumors using methods such as intraoperative stamping and CT-guided localization [6,7]. Traditional localization methods, including preoperative CT-guided marking and intraoperative staining, often result in complications such as dye diffusion, implant displacement, and radiation exposure. These challenges underscore the need for less invasive and more efficient techniques.

Indocyanine green (ICG), a water-soluble dye that emits fluorescence under near-infrared light, presents a significant advancement in tumor localization and lung segmentation during surgeries. Its application in thoracoscopy is gaining traction due to the enhanced precision it offers in surgical procedures, without the negative side effects typically associated with traditional techniques [8,9,10,11]. Notably, while intravenous ICG administration is commonly employed in thoracic surgeries to delineate resection margins, the specific use of intravenous ICG for tumor identification has been explored primarily in clinical trials, with limited application in routine clinical settings [12,13]. Moreover, the percutaneous or transbronchial methods of ICG marking, though practiced in a select few institutions, are complex procedures marked by concerns over their invasiveness and potential complications [3,5].

Given these considerations, the spectral measurement system we propose, designed for intravenous application, stands to not only enhance the existing utilization of ICG in thoracoscopic surgeries but also address the limitations and challenges associated with current ICG marking techniques [14].

We developed an innovative ICG fluorescence spectroscopy system designed for precise tumor localization [15]. This system can measure the spectrum of near-infrared light emitted by ICG in tumor tissues, offering a new dimension to surgical navigation. The introduction of a fluorescence spectrum measurement system that detects ICG emissions invisible to the naked eye marks a significant advancement. This technology has the potential to revolutionize surgical navigation by accurately identifying tumors deep within the lung tissue, thereby enhancing the precision of thoracoscopic surgery. As this technology awaits clinical application, it underscores the ongoing evolution towards more precise and minimally invasive surgical interventions for lung cancer treatment. 

This study investigates the application of ICG and fluorescence spectral measurements in thoracoscopic surgery to enhance the accuracy of visual tumor localization. Utilizing a novel ICG fluorescence spectroscopy system, we aim to determine the optimal ICG dose and administration method for accurate tumor identification during lung resection surgeries.

## 2. Experimental Design and Detailed Procedure

### Study Design

This is a single-center, open, dose-comparison control, interventional clinical trial.

The trial has been registered in the jRCT Clinical Trials Registry under the code jRCTs011230037 (https://jrct.niph.go.jp/en-latest-detail/jRCTs011230037) (accessed on 28 March 2024).

The detailed procedure is described in Figure 1.

The data obtained will be used for analysis until 1 August 2026. Subsequently, the data will be stored and may be used for future studies.
(1)Dissolve indocyanine green (ICG) (Diagnogreen^®^, Daiichi-Sankyo, Tokyo, Japan) in 5 mL to 10 mL of the attached injection solution, dilute it with 200 mL of saline solution to make a total volume of approximately 205–250 mL, and inject it intravenously the day before surgery (approximately 24 h (window of 22 to 26 h) before the surgery) over approximately 1–2 h. The dose should be in the range of 0.5 mg/kg to 5.0 mg/kg. The initial dose should be 0.5 mg/kg. If the positive diagnosis rate is low, the dosage should be increased at 1.0 mg/kg intervals to a maximum of 5.0 mg/kg to determine the optimal dosage for detecting tumors.(2)The probe of the fluorescence spectroscopy system is used in the thoracic cavity, and the ICG is excited by the excitation light source when the probe is applied to the lung tumor to observe whether a fluorescence wavelength is detected. Measure the distance from the surface of the lung to the probe, and the distance from the surface of the lung to the tumor.(3)Pulmonary lobectomy and more extensive surgical approaches are performed as usual in our department. The probe of the fluorescence spectroscopy system is applied to the resected lung removed from the thoracic cavity to generate ICG fluorescence using an excitation light source to determine if the fluorescence wavelength is detected.(4)Postoperative observation of patient for 1 to 2 weeks.

## 3. Materials and Equipment

### 3.1. Fluorescence Spectroscopy System

The near-infrared fluorescence spectroscopy system developed by the Advantest Corporation in Tokyo (https://www.advantest.com/) (accessed on 28 March 2024) is designed to detect the spectrum of near-infrared light [15]. It received clinical approval for use in the human body, specifically for intraperitoneal administration, in February 2023 (approval number: 30500 BZX00031000). Unlike traditional near-infrared thoracoscopes that rely on visual confirmation of ICG excitation, this advanced system can detect ICG emissions that are invisible to the naked eye by capturing the specific wavelengths emitted by ICG. This capability is particularly valuable for identifying ICG deposits deep beneath the lung surface, where emissions may not be visually detectable.

### 3.2. Lumifinder™ MED7100

Generic name: ICG fluorescence observation device (JMDN code: 71076002).

Classification: controlled medical device (Class II), specified maintenance control medical device.

Medical device approval number: 30500BZZZX00031000.

Dimensions: approx. 390 (W) × 250 (H) × 400 (D) mm.

Weight: 30 kg or less.

Power supply: AC100 v 50/60 Hz.

Power consumption: 170 VA.

Type of protection against electric shock: Class I.

Degree of protection against electric shock: BF-type mounting part.

Operating environment Temperature range: 10–30 °C.

      Humidity range: 30–80%

      Air pressure range: 700~1060 hPa

Scope Insertion diameter: 10 mm.

      Effective length of insertion part: 288.9 mm

      Beam direction: 0°

      Beam angle: 25.4°

Laser Specifications: excitation wavelength—785 ± 10 nm, power—40 mW max.

Guiding light: wavelength—520 ± 10 nm, power—10 mW max.

Laser class: 3R.

### 3.3. Allocation Method

The initial ICG dose will be 0.5 mg/kg. Thereafter, the dose will be increased to 1.0 mg/kg, 2.0 mg/kg, and so on, at 1.0 mg/kg intervals to a maximum of 5.0 mg/kg. In this study, the lowest dose of ICG used for blood flow evaluations during the segmentectomy of the lungs and liver was 0.5 mg/kg. Additionally, for the purpose of a safety evaluation, a minimum dose of 0.5 mg/kg was established. Ten patients will be allocated this in order from the lowest dose, resulting in a total sample size of 60 patients.

The sensitivity and specificity of tumor detection at each dose will be calculated using the following formulae.

Sensititvity = number of cases in which tumors were identified by the fluorescence spectroscopy system/total number of cases in which pathology resulted in a diagnosis of malignancy.

Specificity = number of cases in which no tumor was identified by the fluorescence spectroscopy system/total number of cases in which pathology showed a benign tumor.

The most suitable dose for tumor detection will be determined by taking into consideration both accuracy and dosage. While the maximum dose is 5.0 mg/kg, if the positive detection rate becomes 0.9 or higher at a certain dose, and improvement in the positive detection rate cannot be expected further by increasing the dose thereafter, the dosage should not be increased. 

### 3.4. Endpoints

The primary endpoint of the study is the correct diagnostic rate for tumor localization using an intravenous ICG administration. The secondary endpoints include the measurement of intraoperative ICG fluorescence spectral intensity in lung tumors, the assessment of operability and safety when ICG is used intraperitoneally, the measurement of ICG fluorescence spectral intensity in surgical specimens, the comparison of the spectral intensity in lung tissues during collapse and expansion, the correlation between ICG camera images and fluorescence spectral intensity, and the comparison of fluorescence analysis results with histopathological findings from surgical specimens.

Safety endpoints include the evaluation of adverse events identified after the administration of the study drug.

### 3.5. Study Patients and Eligibility Criteria

Patients attending or being admitted to the Department of Thoracic Surgery, Hokkaido University Hospital, and deemed suitable for surgery by the department will be enrolled in this study.

The eligibility criteria are shown in Table 1.

### 3.6. Target Number of Patients

A maximum of 60 patients will be enrolled, including 10 patients per dosage level.

### 3.7. Basis for Setting the Target Number of Cases

We defined the positive diagnosis rate as the proportion of accurately identified malignant or benign cases to the total number of cases examined. Anticipating a positive diagnosis rate of 0.9, we determined that a minimum of ten cases per dosage level will be required. This sample size ensures a 70% probability of estimating the positive diagnosis rate with a margin of error not exceeding 0.4, within a 95% confidence interval. Thus, given the inclusion of up to six different dosage levels in our study, the total sample size could reach up to 60 cases.

### 3.8. Statistical Analysis Methods

The data obtained will be summarized using summary statistics (mean, standard deviation, median, and interquartile range) or frequencies and percentages, both overall and stratified by dose. For each dose, estimates of the positive diagnostic rate, sensitivity, specificity, and 95% confidence intervals will be calculated based on the Wilson score method. In addition, the mean and 95% confidence intervals of the numerical values of wavelength intensity in the peak range of the ICG fluorescence wavelengths will be calculated and compared between doses using an analysis of variance. Subgroups, such as expanded vs. collapsed lungs and non-tumor vs. tumor areas, will also be examined. Adverse events will be presented as the number of occurrences per dose.

### 3.9. Anticipated Benefits and Disadvantages (Burdens and Risks)

Anticipated Benefits: The use of ICG and fluorescence spectroscopy during surgery may uncover lesions, such as pleural seeding, not identified in preoperative assessments. These discoveries could inform real-time adjustments to surgical techniques or treatment plans, potentially avoiding unnecessary procedures and optimizing patient care through integrated therapies, including chemotherapy. Furthermore, the insights gained could advance the development of comprehensive, multidisciplinary treatment approaches.

Anticipated Disadvantages (Burdens and Risks): Adverse reactions, as detailed in the ICG package insert, may include shock, anaphylaxis, and hypersensitivity reactions (e.g., nausea, urticaria, fever), with a frequency of less than 0.1%. ICG administration intravenously may also cause discomfort at the injection site.

While the Lumifinder system is approved for laparoscopic use, it is not specifically indicated for thoracoscopic procedures. However, its intraperitoneal application suggests a low risk of adverse events during thoracoscopic use. The application of Lumifinder’s probe to lung tumors may extend surgical times by up to 15 min, although this is not expected to delay the overall duration of surgery or affect patient recovery. Participation in this study will not increase the volume or frequency of blood tests.

Comprehensive Evaluation: This study does not promise direct benefits to its participants. The main concerns are discomfort from the IV insertion and the rare side effects of ICG. These factors are unlikely to impact overall treatment outcomes. To mitigate risks, individuals with allergies to ICG, iodine, or contrast agents will be excluded from participation.

### 3.10. The Handling of Adverse Events

Treatment of Research Subjects in the Event of an Adverse Event: An adverse event is defined as any undesirable or unintended injury, illness, or symptom (including abnormal laboratory findings) that occurs in a research subject, irrespective of its causal relationship with the research activities. Upon observing an adverse event, the responsible researcher must promptly initiate the appropriate actions and treatment. Details such as the name of the adverse event, date of onset, severity (mild, moderate, severe), outcome, and its relationship to the study (related or unrelated), among other information, must be meticulously recorded. In cases where the study intervention is halted or a specific treatment for the adverse event is required, the research subject will be duly informed.

Reporting of Serious Adverse Events: Serious adverse events are characterized as 1. Resulting in death; 2. Being life-threatening; 3. Necessitating or prolonging hospitalization; 4. Leading to persistent or significant disability or dysfunction; or 5. Resulting in congenital anomalies or birth defects in offspring.

Upon the identification of a serious adverse event, the principal investigator (or an authorized delegate) is obliged to undertake necessary actions, including informing the affected research subjects, and to report the event expeditiously to the overseeing principal investigator. This lead investigator is then responsible for promptly notifying the administrative heads of the medical institution, taking suitable measures, and ensuring that all research personnel and relevant parties are informed about the incident.

The Reporting of Significant Adverse Events:

Significant adverse events include, but are not limited to, conditions like eczema, nausea, vomiting, etc., associated with the administration of ICG and any adverse event resulting in the discontinuation of the study intervention.

Investigators must report incidents qualifying as significant adverse events with the same urgency as serious adverse events.

Documentation of Other Adverse Events: all other adverse events, regardless of severity, must be accurately documented in the medical records by the principal investigator or a designated sub-investigator.

### 3.11. Discontinuation or Termination of Research

Non-participation Treatment: should a patient choose not to participate in the study, they will receive standard pulmonary resection and postoperative care, excluding any research-related evaluations. Discontinuation by Research Subjects: In the event of the following circumstances, the principal investigator will immediately cease the research for the affected subject and ensure their safety through appropriate follow-up and medical care, contingent on the subject’s willingness to cooperate. The extent of the medical interventions at discontinuation will be tailored to the individual’s medical needs and their consent. Data from these subjects will be promptly compiled. If a subject becomes unreachable during the study, efforts will be made to contact them and assess their health status remotely if necessary. Adverse events will be managed as outlined in a separate section. The reasons and dates of discontinuation will be recorded in the EDC system, with the discontinuation date marked as the day the principal investigator decides to terminate the subject’s participation.

Discontinuation Criteria: 1. Subject requests to alter or stop the treatment. 2. Observation of an adverse event that makes further participation inadvisable. 3. Occurrence of death or a potentially fatal illness. 4. Worsening of the primary disease. 5. Discovery of subject’s pregnancy. 6. Deviation from inclusion criteria or breach of exclusion criteria by the subject. 7. Subject’s failure to attend hospital visits. 8. Any other situation where continuation is deemed inappropriate by the investigator(s).

Rationale: Criterion 1 is established based on ethical considerations. Criteria 2 through 5 are set for safety reasons. Criteria 6 and 7 reflect the practical challenges in continuing the research under certain conditions.

### 3.12. Discontinuation of the Entire Study

The principal investigator may consider halting the entire study under any of the following conditions. In the event of such a discontinuation, the investigator is required to inform the Review Committee and the Minister of Health, Labour and Welfare, using the designated form, within 10 days. Additionally, the investigator must promptly notify the administrative head of the medical institution conducting the study and all participating research subjects, ensuring their safety through necessary examinations and interventions.

Conditions for Study Discontinuation: 1. Discovery of information that challenges the ethical integrity or scientific validity of the study or impacts its continuation. 2. Acquisition of facts or information that may compromise the proper conduct or credibility of the research findings. 3. Determination that the research risks outweigh the expected benefits, or the receipt of information suggesting that the research will not yield meaningful results. 4. The receipt of significant data concerning the quality, safety, or effectiveness of the research equipment that necessitates the termination of the study.

Furthermore, if there is a grave breach of the Clinical Research Act, this research protocol, or the relevant agreements by the medical institution or its staff, or if the proper conduct of the research becomes untenable, the principal investigator may request that the institution ceases or pauses the research. In cases where the study is halted or suspended due to reasons originating within the implementing institution, such as recommendations or directives from the review committee to stop the research or advisories from the Review Committee to alter the research plan in ways that are impractical to implement, the investigator will swiftly inform all participating research subjects and take the necessary steps to ensure their safety.

### 3.13. Termination of Research

A study is considered terminated upon the completion of all the following criteria: 1. All study subjects have been enrolled, and the observation period has concluded. 2. The primary endpoint report, along with a comprehensive summary report and an abridged version of this summary, have been prepared. 3. The primary endpoint report has been submitted to the Minister of Health, Labour and Welfare. 4. A concise version of the summary report, alongside the research plan and statistical analysis plan, has been submitted to the Minister of Health, Labour and Welfare. 5. The primary endpoint report, the full summary report, and its abridged version have been provided to the administrator of the medical institution conducting the study. 6. A summary of the research findings has been registered in the Japan Registry of Clinical Trials (jRCT). 7. The publication of the research results has been reported to the administrator of the implementing medical institution.

### 3.14. Post-Research Actions

Following the completion of the study, research subjects will be offered the medical care considered most suitable, taking into account the findings from this research.

### 3.15. Data Collection

The Electronic Data Capture (EDC) system will be utilized to document all necessary information pertinent to this study. It is the responsibility of the principal investigator and the administrator of the executing medical institution to verify the accuracy and completeness of the data recorded in the case report forms and any additional reports.

Case Report Form Preparation and Data Management: The principal investigator will create a comprehensive guide titled “Guide to Completing Case Report Forms”, detailing the procedures and guidelines for accurately filling out these forms. Individuals responsible for data entry will follow this guide to input information into the EDC system, a process referred to as “Case Report Form Preparation: Entry into the EDC System”. Data entry into the EDC system will be conducted by the principal investigator or designated research associates for all participants enrolled in the study. This process will continue as needed throughout the duration of the study until its conclusion. Upon data entry, each participant will be assigned a unique identifier (research subject identification code) that is not linked to their personal information. A corresponding table will be created to match the data with the respective participants; however, this table will exclude any identifiable information. The table will be securely stored at the implementing medical institution, ensuring the confidentiality of participant information.

### 3.16. Monitoring

The monitoring of this research shall be conducted in accordance with the Procedures for Monitoring. Furthermore, the principal investigator and the implementing institution are required to make all research-related records of the research subjects available for direct inspection upon request by the monitoring personnel. The person in charge of monitoring shall verify that this research is being conducted in compliance with the Clinical Research Act, relevant notifications, and the research protocol.

## 4. Expected Results and Discussion

Our research is dedicated to assessing the capabilities of an advanced ICG fluorescence spectroscopy system to enhance tumor localization during thoracoscopic surgery. Our prior investigations into this state-of-the-art system have shown its proficiency in identifying tumors located deeper than the lung surface—regions that conventional ICG cameras struggle to reveal [15]. A particular challenge has been detecting lung tumors smaller than 5 mm and situated more than 20 mm beneath the surface, which are often not visible with standard ICG cameras or to the naked eye. Based on previous studies, to achieve this objective, we intravenously administer ICG and image it at a later time (24 h after delivery, during the second window) [12]. The feasibility of NIR imaging in the second window is due to the dye’s accumulation in tissues with abnormally leaky capillaries and/or increased pressure gradients through the EPR effect.

The novel spectral system we have developed for our current study distinguishes itself by detecting NIR light wavelengths through specialized instruments, potentially identifying lesions undetectable when using traditional ICG cameras [15]. Our preliminary results have shown that while conventional ICG cameras falter in identifying pseudo-tumors concealed by more than 10 mm of tissue or tissue substitute, our spectral analysis device can discern spectra obscured by even thicker layers. These findings highlight the crucial roles of both the distance between the tumor and the probe and the density of the tissue between them in the detection of ICG wavelengths.

We are poised to launch a clinical trial to appraise the system’s utility in surgical navigation, focusing particularly on the fluorescence spectrum of lung cancer and metastatic lung tumors. This study will encompass patients slated for lung resection due to these conditions, employing the ICG system to intraoperatively examine the fluorescence spectrum of their tumors and to further analyze the excised tissue postoperatively. The surgical procedures will conform to standard lung resection protocols.

An intriguing benefit of using an intraoperative ICG camera is its potential to uncover additional lesions, like pleural seeding, that might not have been detected during preoperative assessments. The discovery of such lesions could necessitate immediate adjustments in surgical tactics and treatment plans, allowing these insights to be integrated into comprehensive patient care seamlessly.

## 5. Conclusions

Our study aims to underscores the potential of fluorescence spectroscopy to revolutionize the localization of lung tumors during thoracoscopic surgeries. By demonstrating the effectiveness of this system, we aim to establish a novel and non-invasive method for tumor identification with better reliability than that of conventional techniques. The practical application of this system holds promise for standardizing lung tumor localization across various medical facilities, making advanced surgical navigation accessible and dependable, regardless of the setting. This advancement not only signifies a leap forward in thoracic surgery but also exemplifies our commitment to enhancing patient outcomes through innovative technological integration.

## Figures and Tables

**Figure 1 jcm-13-02807-f001:**
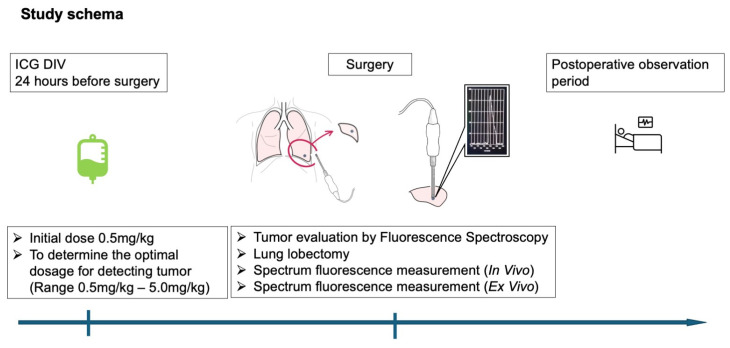
Detailed procedure.

**Table 1 jcm-13-02807-t001:** Eligibility criteria.

Inclusion Criteria	Exclusion Criteria
(i) Male and female patients aged 20 years or older at the time of consent	(i) Patients under 20 years of age at the time of consent
(ii) Patients with a preoperative diagnosis of primary lung cancer (or suspected primary lung cancer) or metastatic lung tumor (or suspected metastatic tumor)	(ii) Patients with a history of an allergy to ICG, iodine, or contrast media (because ICG contains a small amount of iodine)
(iii) Patients with a tumor diameter of ≥1 cm evident on a CT scan and those with a tumor located within a 2 cm depth from the lung surface	(iii) Patients with interstitial pneumonia or pulmonary fibrosis evident on chest CT
(iv) Patients scheduled for a lobectomy or more extensive surgical procedures	(iv) Pregnant or lactating mothers
(v) Patients who have received a full explanation of their participation in this study and have given their free and voluntary written consent based on their full understanding	(v) Patients who are abstinent or unable to use effective contraceptive methods during the period of participation in the study
	(vi) Patients who have difficulty with their oral intake
	(vii) Patients with obvious hepatic or cardiac dysfunction
	(viii) Patients with dementia or other conditions that require surrogate consent

## Data Availability

The data underlying this article will be shared upon reasonable request to the corresponding author.

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
