# Peer review of "Dose Optimization of Intravenous Indocyanine Green for Malignant Lung Tumor Localization"

_jcm, 2024, doi:10.3390/jcm13102807_

Round 1

Reviewer 1 Report

Comments and Suggestions for Authors

First, the concentration of indocyanine green, including how the initial
concentration is determined.

Secondly, if indocyanine green injection cannot reveal the location of
pulmonary nodules, especially small ground-glass nodules, how should we
perform subsequent surgery?

Third, are there differences in fluorescence between tumors of different
pathological types?

Finally, the distance between the nodule and the lung surface should be
considered. 

Comments on the Quality of English Language

The quality of the English language is ok

Author Response

Dear Reviewer,

I would like to thank you for reviewing our manuscript. Please find below our responses to your questions and suggestions.

  1. First, the concentration of indocyanine green, including how the initial concentration is determined.

Answer:

According to the Food and Drug Administration approvals and previous reports [14], the safety of ICG has been established for systemic administration at a dose of 0.5 mg/kg intravenously. In this study, the lowest dose of ICG used for evaluating blood flow during segmentectomy of the lungs and liver was 0.5 mg/kg. For safety evaluations, a minimum dose of 0.5 mg/kg was similarly established.

The following text has been added in Page 4, Line 146

In this study, the lowest dose of ICG used for blood flow evaluation during segmentectomy of the lungs and liver was 0.5 mg/kg. Additionally, for the purpose of safety evaluation, a minimum dose of 0.5 mg/kg was established.

2. Secondly, if indocyanine green injection cannot reveal the location of pulmonary nodules, especially small ground-glass nodules, how should we perform subsequent surgery?

Answer:

As you mentioned, small tumors, especially small ground-glass nodules, are likely to be difficult to identify with ICG. Therefore, we only enrolled patients who were scheduled to undergo lobectomy or more extensive procedures in this study. The inclusion criteria were specifically designed to include cases scheduled for lobectomy.

The following text has been added in inclusion criteria in Page 3, Line 105 and Table 1

Patients scheduled for a lobectomy or more extensive surgical procedures.

  1. Third, are there differences in fluorescence between tumors of different pathological types?

Answer:

Thank you for your question. We believe that there are differences in fluorescence among tumors of various pathological types. In this study, we will also examine differences in fluorescence intensity across different pathologies.

Perdina et al.[18] mentioned in their previous study that they have found success using ICG as a tumor mapping agent for various solid tumor histologies. Their experiences with sarcomatous pulmonary metastases are particularly encouraging.

4. Finally, the distance between the nodule and the lung surface should be considered. 

Thank you for pointing that out.

As mentioned in the discussion section, the location of the tumor relative to the lung surface is crucial.

Our research is dedicated to assessing the capabilities of an advanced ICG fluorescence spectroscopy system in enhancing tumor localization during thoracoscopic surgery. Our prior investigations have demonstrated this state-of-the-art system’s proficiency in identifying tumors located deeper than the lung surface—areas where conventional ICG cameras typically falter. A particular challenge has been detecting lung tumors smaller than 5 mm and situated more than 20 mm beneath the surface, which often remain invisible to standard ICG cameras and the naked eye.

Additionally, the distance between the probe and the lung surface should also be considered.

The following text has been added to Page 3, Line 103

Measure the distance from the surface of the lung to the probe, and the distance from the surface of the lung to the tumor.

Reviewer 2 Report

Comments and Suggestions for Authors

I would like to congratulate the authors on embarking on an interesting and much-needed field of research, which currently lacks consensus.

I would, however, like to ask you a couple of questions.

1) From reading of the literature it seems that the half-life of Indocyanine green is quite short, it may be helpful to explain how injection 24 hours prior to the procedure would be helpful in adequate tumor localization. One could make the assumption that the nature of the solution/emulsion, you are using provides sufficient visualization at 24 hours, but this does not seem to be clearly apparent to the reader.

2) I see that you have proposed a gradual dose escalation, depending upon the degree of tumor visualization. I wonder if it would not be a better approach to have pre-specified dosing that the operator is blinded to and then reporting the operator satisfaction, as well as the diagnostic accuracy in quartiles?

Author Response

Dear Reviewer,

I would like to thank you for reviewing our manuscript. Please find below our responses to your questions and suggestions.

  1. From reading of the literature it seems that the half-life of Indocyanine green is quite short, it may be helpful to explain how injection 24 hours prior to the procedure would be helpful in adequate tumor localization. One could make the assumption that the nature of the solution/emulsion, you are using provides sufficient visualization at 24 hours, but this does not seem to be clearly apparent to the reader.

Thank you for pointing this out. Predina et al.[18] mentioned in their previous study that ICG has traditionally been used to assess tissue perfusion. However, we have modified the delivery parameters to deploy ICG as a tumor mapping agent. To accomplish this, we intravenously administer ICG at a much higher dose (5 mg/kg) and image at a later time (24 hours post-delivery, during the so-called second window). We refer to this imaging approach as "Tumor Glow." The feasibility of NIR imaging in the second window is supported by the dye's accumulation in tissues with abnormally leaky capillaries and/or increased pressure gradients through the EPR effect.

Based on the results of their study, we too have adopted the approach of imaging 24 hours after delivery.

Additionally, the following text has been added Page 9, Line 353

Based on previous studies, to achieve this objective, we intravenously administer ICG and image at a later time (24 hours after delivery, during the second window). The feasibility of NIR imaging in the second window is due to the dye's accumulation in tissues with abnormally leaky capillaries and/or increased pressure gradients through the EPR effect.

2) I see that you have proposed a gradual dose escalation, depending upon the degree of tumor visualization. I wonder if it would not be a better approach to have pre-specified dosing that the operator is blinded to and then reporting the operator satisfaction, as well as the diagnostic accuracy in quartiles?

Thank you for pointing this out. Based on a previous report, administering a 5 mg/kg dose of ICG intravenously has shown better results for tumor identification[14,18]. However, this dosage has never been used in Japanese patients. Safety should be considered carefully, starting with small quantities and progressing in stages. Therefore, we will study the method of ICG intravenous administration and the optimal dosage as a means of tumor identification during lung resection.

Reviewer 3 Report

Comments and Suggestions for Authors

Minor point - current status of indocyanine fluorescence is very positive - but with all of the limitations you mention - consider this reference

Cui F, Liu J, Du M, Fan J, Fu J, Geng Q, He M, Hu J, Li B, Li S, Li X, Liao YD, Lin L, Liu F, Liu J, Lv J, Pu Q, Tan L, Tian H, Wang M, Wang T, Wei L, Xu C, Xu S, Xu S, Yang H, Yu BT, Yu G, Yu Z, Lee CY, Pompeo E, Azari F, Igai H, Kim HK, Andolfi M, Hamaji M, Bassi M, Karenovics W, Yutaka Y, Shimada Y, Sakao Y, Sihoe ADL, Zhang Y, Zhang Z, Zhao J, Zhong W, Zhu Y, He J. Expert consensus on indocyanine green fluorescence imaging for thoracoscopic lung resection (The Version 2022). Transl Lung Cancer Res. 2022 Nov;11(11):2318-2331.

The dose escalation should be a little more specific - first dose escalation should be from 0.5 to 1 - otherwise the max dose would be 4.5 and not 5.

Will there be an oversight or data monitoring committee? If not will this be the responsibility of the lead investigator?

Comments on the Quality of English Language

The English is just a little clunky in parts. Not an absolute issue, but could be improved

ICG should be dissolved in the provided solution or distilled water and administered intravenously approximately 24 h before surgery

ICG is dissolved........approximately 24h ... what was the window? 22-26hrs?

Author Response

Dear Reviewer,

I would like to thank you for reviewing our manuscript. Please find below our responses to your questions and suggestions.

  1. Minor point - current status of indocyanine fluorescence is very positive - but with all of the limitations you mention - consider this reference

Cui F, Liu J, Du M, Fan J, Fu J, Geng Q, He M, Hu J, Li B, Li S, Li X, Liao YD, Lin L, Liu F, Liu J, Lv J, Pu Q, Tan L, Tian H, Wang M, Wang T, Wei L, Xu C, Xu S, Xu S, Yang H, Yu BT, Yu G, Yu Z, Lee CY, Pompeo E, Azari F, Igai H, Kim HK, Andolfi M, Hamaji M, Bassi M, Karenovics W, Yutaka Y, Shimada Y, Sakao Y, Sihoe ADL, Zhang Y, Zhang Z, Zhao J, Zhong W, Zhu Y, He J. Expert consensus on indocyanine green fluorescence imaging for thoracoscopic lung resection (The Version 2022). Transl Lung Cancer Res. 2022 Nov;11(11):2318-2331.

Thank you for pointing this out. I have also read your paper and found it to be very well-organized and informative regarding ICG. I will definitely add it to my references as [20].

  1. The dose escalation should be a little more specific - first dose escalation should be from 0.5 to 1 - otherwise the max dose would be 4.5 and not 5.

According to the Food and Drug Administration-approved guidelines in Japan and previous reports[14,18], the safety of ICG has been established for systemic administration at a dose of 0.5 mg/kg intravenously. In this study, the lowest dose of ICG used for the segmentectomy of the lungs and evaluation of liver blood flow was also 0.5 mg/kg, to ensure safety. Based on previous reports, a dose of 5 mg/kg intravenously is considered better for tumor identification. However, this dosage has never been used in Japanese patients. Safety should be considered in small quantities and in stages. Therefore, the method of ICG intravenous administration and the optimal dose for tumor identification during lung resection will be further studied.

  1. Will there be an oversight or data monitoring committee? If not will this be the responsibility of the lead investigator?

Thank you for pointing this out. We have a monitoring committee for this study.
The following sentence has been added to the article under the Page 9 Line 338

Monitoring:

Monitoring of this research shall be conducted in accordance with the Procedures for Monitoring. Furthermore, the principal investigator and the implementing institution are required to make all research-related records of the research subjects available for direct inspection upon request by the monitoring personnel. The person in charge of monitoring shall verify that this research is being conducted in compliance with the Clinical Research Act, relevant notifications, and the research protocol.

  1. The English is just a little clunky in parts. Not an absolute issue, but could be improved.

Thank you for pointing this out, we have asked for a Native check.

  1. ICG should be dissolved in the provided solution or distilled water and administered intravenously approximately 24 h before surgery.

ICG is dissolved........approximately 24h ... what was the window? 22-26hrs?

We have set the ICG injection time to approximately 24 hours before the initial skin incision. However, it is important to establish a specific window period for this procedure. Based on your suggestion, I think setting the window from 22 to 26 hours before surgery is reasonable.

I have added this sentence to the Page 3 Line 98

"approximately 24 hours (window 22 to 26 hours) before the surgery."